# Photocatalysis and Photodynamic Therapy in Diabetic Foot Ulcers (DFUs) Care: A Novel Approach to Infection Control and Tissue Regeneration

**DOI:** 10.3390/molecules30112323

**Published:** 2025-05-26

**Authors:** Paweł Mikziński, Karolina Kraus, Rafał Seredyński, Jarosław Widelski, Emil Paluch

**Affiliations:** 1Faculty of Medicine, Wroclaw Medical University, Wyb. Pasteura 1, 50-376 Wroclaw, Poland; pawel.mikzinski@student.umw.edu.pl (P.M.); karolina.kraus@student.umw.edu.pl (K.K.); 2Department of Physiology and Pathophysiology, Wroclaw Medical University, Tytusa Chałubińskiego 10, 50-368 Wroclaw, Poland; rafal.seredynski@umw.edu.pl; 3Department of Pharmacognosy with Medicinal Plants Garden, Lublin Medical University, 20-093 Lublin, Poland; jaroslaw.widelski@umlub.pl; 4Department of Microbiology, Faculty of Medicine, Wroclaw Medical University, Tytusa Chalubinskiego 4, 50-376 Wroclaw, Poland

**Keywords:** diabetic foot ulcers, diabetic wound, photocatalysis, photosensitizer, tissue regeneration, antibacterial photodynamic therapy, nanoparticles, liposomes, biofilm prevention, oxidative stress

## Abstract

Photocatalysis and photodynamic therapy have been increasingly used in the management of diabetic foot ulcers (DFUs), and their integration into increasingly innovative treatment protocols enables effective infection control. Advanced techniques such as antibacterial photodynamic therapy (aPDT), liposomal photocatalytic carriers, nanoparticles, and nanomotors—used alone, in combination, or with the addition of antibiotics, lysozyme, or phage enzymes—offer promising solutions for wound treatment. These approaches are particularly effective even in the presence of comorbidities such as angiopathies, neuropathies, and immune system disorders, which are common among diabetic patients. Notably, the use of combination therapies holds great potential for addressing challenges within diabetic foot ulcers, including hypoxia, poor circulation, high glucose levels, increased oxidative stress, and rapid biofilm formation—factors that significantly hinder wound healing in diabetic patients. The integration of modern therapeutic strategies is essential for effective clinical practice, starting with halting infection progression, ensuring its effective eradication, and promoting proper tissue regeneration, especially considering that, according to the WHO, 830 million people worldwide suffer from diabetes.

## 1. Introduction

### 1.1. Diabetic Foot Ulcers (DFUs)—Devastating Complication of Diabetes Mellitus

Diabetes mellitus (DM) is a metabolic disorder characterized by elevated blood glucose levels. It is one of the most prevalent chronic diseases in the world. In the United States, it remains the seventh leading cause of death [1]. Diabetic wounds occur as a late complication of vascular, neurological, and biomechanical issues caused by diabetes. They typically affect the lower limbs, particularly the feet, in the form of diabetic foot ulcers. Each year, around 18.6 million people globally, including 1.6 million in the United States, develop a diabetic foot ulcer (DFU). These ulcers are a leading cause of lower extremity amputations in individuals with diabetes, accounting for 80% of such cases, and are associated with a higher risk of mortality [2].

### 1.2. Pathophysiology of Diabetic Foot Ulcers

In the case of diabetic foot ulcers (DFUs), the triad of neuropathy, arterial occlusive disease, and trauma with secondary infection is typically responsible for their development. Diabetic neuropathy affects motor, sensory, and autonomic functions, leading to foot deformities, sensory loss, and impaired wound healing. Autonomic neuronal dysfunction, in particular, reduces sweat production, increasing the foot’s susceptibility to dryness and skin cracking.

Arterial occlusive disease in diabetes leads to endothelial damage, atherosclerosis, and impaired perfusion of the feet. Diabetes mellitus (DM) significantly increases the risk of arterial occlusive disease, with patients exhibiting more than twice the prevalence compared to the general population. These factors increase the likelihood of unnoticed trauma due to sensory loss or other diabetes-related complications [3,4].

Diabetes also impairs wound healing, and chronic DFUs often stall at various stages of the healing process. In the early phase, dysregulated neutrophil extracellular traps (NETs) trigger prolonged inflammation, while excessive cytokine production and the accumulation of advanced glycation end-products (AGEs) activate further inflammatory pathways. AGEs disrupt the extracellular matrix (ECM), hindering collagen production and impairing tissue repair.

Angiogenesis is impaired in DFUs due to reduced levels of angiogenic factors such as vascular endothelial growth factor (VEGF) and fibroblast growth factor-2 (FGF-2), along with dysfunctional endothelial progenitor cells (EPCs). AGE accumulation further compromises EPC function, slowing down blood vessel formation and delaying wound healing [5].

The wounds caused by the initial trauma are susceptible to infection due to the prolonged and impaired healing processes. Combined with the compromised immune response seen in diabetic patients, these infections can be particularly dangerous. They can spread rapidly to surrounding tissues, initially causing cellulitis and potentially progressing to more severe complications, such as osteomyelitis or necrotizing fasciitis, significantly increasing the risk of disabling amputations [6,7]. DFUs’ pathophysiology is shown in Figure 1.

### 1.3. Photocatalysis and Photodynamic Therapy as Potential Clinical Approaches for the Management and Treatment of Diabetic Foot Ulcers

Managing and treating patients with diabetic ulcers remains a significant clinical challenge. The complexity of wound healing in diabetic patients often leads to severe complications, including life-threatening infections and a high risk of amputations. As the prevalence of diabetes continues to escalate globally, the medical community is urgently seeking more effective therapeutic strategies to address this critical issue.

Among the emerging treatment modalities, photocatalysis has gained attention due to its potential to enhance wound healing through the generation of reactive oxygen species (ROS), which possess antimicrobial and regenerative properties. Alongside photocatalysis, photodynamic therapy (PDT) has also emerged as a promising technique. PDT utilizes photosensitizers activated by specific wavelengths of light to produce cytotoxic reactive species, targeting microbial cells and promoting tissue regeneration.

Photocatalysis and photodynamic therapy share a common foundation in their reliance on photoactivation to generate reactive oxygen species that exert therapeutic effects. Despite their similarities, each approach has distinct mechanisms and applications, which warrant thorough exploration. In this study, we propose an integrative approach that leverages the synergistic potential of these photo-based therapies, aiming to optimize the treatment outcomes for diabetic ulcers.

### 1.4. Photocatalysis

The field of photocatalysis focuses on using photon energy to trigger chemical reactions on non-adsorbing substrates through mechanisms such as single electron transfer, energy transfer, or atom transfer. The efficiency of these processes depends on the ability of a light-absorbing metal complex, organic molecule, or other substance—known as a photocatalyst (PC)—to facilitate these transformations. The term photocatalyst is derived from two components: photo, referring to photons, and catalyst, a substance that alters the rate of a chemical reaction without being consumed. Consequently, photocatalysts are materials that modulate the reaction kinetics upon exposure to light. Photocatalysts are intentionally applied to lower the activation energy of the reactions they catalyze, enabling their efficient and controlled progression to achieve the desired outcome. Photoredox methods rely on photocatalysts, which have the unique capability of acting as both oxidizing and reducing agents when activated [8].

Photocatalysis is utilized in the treatment of diabetic wounds due to its antibacterial, anti-inflammatory, and tissue-regenerative properties, which are compared in Figure 2. Its therapeutic mechanism is based on the activation of a photocatalyst, such as titanium dioxide (TiO_2_) nanoparticles, upon exposure to light of a specific wavelength. This activation can lead to the production of ROS, which regulate the wound-healing process by influencing inflammation, cell proliferation, angiogenesis, granulation tissue formation, and extracellular matrix production [9]. The term ROS refers to molecules that contain O_2_, but have been reduced by added electrons, transforming them into highly reactive, radical forms. Well-known members of the ROS molecule family include the superoxide anion (·O_2_^−^), hydrogen peroxide (H_2_O_2_), hydroxyl radicals (·OH), and hydroxyl ions (OH^−^). Reactive nitrogen species (RNS) such as nitric oxide (NO) can also be generated during the photocatalysis process and can act as vasodilators and play a role in antioxidant defense in maintaining the proper balance necessary for wound healing [10]. The application of photocatalysis also enables the production of oxygen, which is essential for proper healing and tissue regeneration [11].

### 1.5. Classification of Photocatalysts

In addition to titanium dioxide, a wide variety of photocatalysts have been developed, each with distinct compositions, light absorption properties, and photocatalytic mechanisms. A systematic understanding of the different types of photocatalysts, their key properties, and their modes of action is essential for designing more effective therapeutic strategies. In this section, a comprehensive overview of photocatalysts’ fundamental characteristics will be presented.

#### 1.5.1. Inorganic Photocatalysts—Metal Oxides

Some of the most commonly used photocatalysts that belong to this group are TiO_2_, ZnO, SnO_2_, Cu_2_O, and WO_3_. These materials possess favorable characteristics for photocatalysis, including suitable band gaps and optimal band-edge positions that enable efficient light absorption and the generation of electron–hole pairs (e^−^/h^+^). Furthermore, their large surface areas provide ample reaction sites, while their chemical stability and reusability are advantageous for practical applications. The primary photocatalytic mechanism in metal oxides involves the generation of photoproduced electron–hole pairs, which drive redox reactions. Specifically, the oxidation of OH^−^ leads to the formation of hydroxyl radicals (•OH), and the reduction of O_2_ results in superoxide radicals (O_2_•^−^), both potent reactive species responsible for disinfection and mineralization. While certain metal oxides like Fe_2_O_3_ exhibit visible light activity, their application is often limited by instability and photocorrosion. In contrast, TiO_2_ stands out for its exceptional corrosion resistance, aqueous stability, and robust photocatalytic performance, establishing it as a benchmark photocatalyst for environmental and biomedical applications [12].

When it comes to the safety of those metal oxides, most commonly applied photocatalysts—TiO_2_ and ZnO—are considered as safe (GRAS) by the US Food and Drug Administration [13].

#### 1.5.2. Carbon Dots

Carbon dots (CDs) are versatile photocatalysts and photosensitizers capable of absorbing light across a broad spectrum, including UV, visible, and near-infrared regions. Their non-metallic, inert nature, coupled with their facile synthesis, makes them attractive for the photodegradation of water contaminants. In hybrid materials with semiconductors, C-dots can enhance light absorption and charge separation, thereby improving the efficiency of pollutant removal. Their photocatalytic degradation mechanisms for dyes like methylene blue and Rhodamine-B have been demonstrated [14].

Carbon dots are generally considered biocompatible and exhibit low toxicity, with studies showing minimal cytotoxicity at appropriate concentrations. However, their safety profile depends on factors such as surface modification, particle size, and dosage, with some studies highlighting potential toxicity at higher concentrations or prolonged exposure [15].

#### 1.5.3. Silver-Doped Photocatalysts (Ag-TiO_2_, Ag-ZnO)

Characteristics of Ag-ZnO and Ag-TiO_2_ photocatalysts have been showcased by Bian et al. Those photocatalysts show high efficiency in Rhodamine B (RhB) degradation under UV–vis light. The Ag/ZnO/AgO/TiO_2_ composite outperforms pure TiO_2_ and ZnO, achieving nearly complete degradation of RhB. This enhanced photocatalytic performance is due to the improved charge transfer between the materials, facilitated by heterojunctions that reduce charge recombination. The lower bandgap of the composite allows for better utilization of visible light. The incorporation of silver (Ag) enhances photocatalytic activity by increasing active sites and improving charge separation, leading to faster and more efficient degradation processes [16]. As mentioned before, those metal oxides (TiO_2_ and ZnO) in non silver-doped forms are considered as generally safe. However, silver nanoparticles raise significant safety concerns regarding cytotoxicity and oxidative stress, which will be discussed in detail in later sections.

This section focused on the characterization of photocatalysts that have demonstrated potential for application in diabetic foot ulcer (DFU) therapy. Recent studies have highlighted materials that promote tissue regeneration, exert antibacterial effects, or modulate hyperglycemic conditions in the wound microenvironment [17,18].

### 1.6. The Effect of Photocatalysis on Gram-Positive Versus Gram-Negative Bacteria

Gram-positive bacteria generally exhibit greater susceptibility to the process of photocatalysis compared to Gram-negative bacteria, which is due to fundamental differences in the structure of their cell walls. The key factor determining the effectiveness of bacterial inactivation by ROS generated during photocatalysis is the presence or absence of an outer cell membrane. Gram-positive bacteria have a thick cell wall (20–80 nm) with one layer and more than 50% peptidoglycan content, while Gram-negative bacteria have a thinner wall (10 nm) with two layers and 10–20% peptidoglycan. Gram-positive bacteria have low lipid and lipoprotein content (0–3%), whereas Gram-negative bacteria have high lipid content (58%). Lipopolysaccharides are absent in Gram-positive bacteria but make up 13% in Gram-negative bacteria [19]. Differences in structure and potential sensitivity to the effects of photocatalysis are summarized in Table 1.

Gram-positive bacteria, such as *Staphylococcus aureus* and *Enterococcus faecalis*, possess a thick peptidoglycan layer that serves a structural function, yet it is relatively porous and does not constitute an effective protective barrier against ROS. As a result, reactive hydroxyl radicals (•OH) and peroxides easily penetrate the cell membrane, causing oxidative damage to lipids, proteins, and DNA, leading to the release of proteins and potassium ions from the interior of the bacterial cell [20].

In contrast, Gram-negative bacteria, such as *Escherichia coli* and *Pseudomonas aeruginosa*, exhibit greater resistance to photocatalysis due to the presence of an additional outer membrane composed of lipopolysaccharides (LPSs). This structure acts as a protective barrier, limiting the access of ROS to the internal cell structures. Only after the outer membrane is damaged can reactive oxygen species effectively interact with the cytoplasmic membrane and genetic material of the bacteria. This process is less efficient than in Gram-positive bacteria because it requires more energy and a longer exposure time to the photocatalytic action of the active material, such as TiO_2_ [21].

However, some Gram-positive bacteria have the ability to produce endospores, which provide them with a significant level of resistance against various harsh environmental conditions, ensuring their survival under stressful circumstances. In addition, some bacteria have a unique capability to form a protective layer made of polysaccharides. This extra layer serves as a shield, preventing the bacterium from being degraded by external factors. Furthermore, the susceptibility of different bacterial species to photolysis—damage caused by exposure to light—can vary due to differences in their metabolic pathways and their capacity to regenerate cellular components. Bacteria that possess more advanced antioxidant systems or efficient DNA repair mechanisms are likely to be more resilient and less affected by the harmful effects of photolysis. Additionally, some bacteria, including *Klebsiella oxytoca*, are capable of producing extracellular polymeric substances (EPSs), or even biofilms, which act as protective barriers. These biofilms or EPSs can effectively shield the bacteria from the damaging effects of photolysis, enhancing their overall resistance to environmental stresses [22].

## 2. Novel Types of DFUs Therapy

Photocatalysis and photodynamic therapy have long been utilized in clinical practice, including the treatment of chronic wounds. Recently, improvements have been made to enhance basic applications, such as the traditional interaction between the photosensitizer and the wound. Without careful selection of the applied substance, light type, and treatment protocol, the effectiveness of this approach remained limited [23]. Current research highlights a growing trend toward the use of advanced energy carriers, as well as electron and proton transfer systems, combined with precisely tailored wavelengths. What is more, there is increasing focus on maintaining only the necessary levels of reactive oxygen species (ROS) during therapy, while implementing systems aimed at reducing their excess, with the goal of achieving even better clinical outcomes. Increasingly, these innovations are being combined with other therapeutic methods as part of combination therapy, which can significantly improve outcomes, even when photosensitizers alone, such as Toluidine Blue or Rose Bengal, might not be effective enough, as will be discussed in the following section.

An important aspect we explore is not only the effective eradication of pathogens but also the adaptation of therapy to the specific wound environment, particularly in patients with diabetes. Given the metabolic changes associated with this condition—characterized by glucose-rich and oxygen-deficient DFU environments—optimizing treatment strategies to address these unique challenges is essential. In Figure 3, we present a summary of the methods discussed later in our review.

### 2.1. Antibacterial Photodynamic Therapy

Antibacterial photodynamic therapy (aPDT) is a promising approach to treat antibiotic-resistant bacterial infections. aPDT involves photosensitizing bacteria using exogenous compounds called photosensitizers (PSs). Cell death is induced by lethal oxidative stress caused by irradiating the infected area with light of a resonant wavelength, typically in the visible range (380–780 nm). The light-sensitive PS, present in the bacteria or on their surface, absorbs light and transitions to a singlet state (^1^PS). The excited electrons then undergo intersystem crossing to a longer-lived triplet state (^3^PS), from which reactive oxygen species (ROS) or reactive molecular intermediates are generated. Photochemical reactions in aPDT occur through type I and type II mechanisms, both requiring close proximity between the ^3^PS and substrate. In type I reactions, electron transfer from the ^3^PS to the substrate generates radicals, often involving oxygen to produce superoxide anion (O_2_^•‒^). While O_2_^•‒^ is generally harmless in biological environments, it can lead to the generation of reactive oxygen species (ROS). Type II reactions involve the direct interaction of the excited PS with oxygen to form singlet oxygen (^1^O_2_) via energy transfer. Both mechanisms likely occur together during aPDT [24] as shown in Figure 4.

aPDT has several advantages over antibiotics. One of its key benefits is its targeted action, as photosensitizers (PSs) are predominantly absorbed by the target cells rather than non-target ones. Additionally, PSs remain pharmacodynamically inactive without irradiation, and the treatment is confined to the illuminated infected area. This specificity significantly reduces systemic toxicity outside the treated zone. Another important advantage is that aPDT does not induce resistance, as repeated treatments have not been found to select for resistant bacterial strains [25]. The interval between the administration of the photosensitizer (PS) and photodynamic therapy (PDT) is too short for bacteria to develop resistance. The severe cellular damage induced by PDT inhibits the ability of bacteria to develop and transmit adaptive resistance mechanisms across generations. One of the most valuable advantages of aPDT is its multifaceted mechanism of action, which differs significantly from conventional antibiotics that typically target a single bacterial component. The reactive oxygen species (ROS) generated during aPDT induce extensive damage to multiple cellular structures and disrupt various metabolic pathways, creating substantial barriers to the development of resistance mechanisms [26,27].

The effectiveness of aPDT largely depends on the optimal combination of photosensitizer (PS) and light. An ideal PS should have high phototoxicity, low dark toxicity, high quantum yield of ^1^O_2_ or free radicals, preferential binding to bacteria over mammalian cells, suitable pharmacokinetics, and accumulation in bacteria or attachment to the bacterial cell envelope. PS binding and uptake depend on the bacterial species. Gram-positive bacteria are more susceptible to anionic and neutral PSs due to their thick, porous peptidoglycan layer, while Gram-negative bacteria are less likely to uptake PSs because of their additional outer membrane and lipopolysaccharide barrier [28]. For optimal aPDT, cationic PSs are preferred for both Gram-positive and Gram-negative bacteria. Cationic derivatives of phenothiazines, phthalocyanines, and porphyrins have been shown to significantly increase phototoxicity in both types of bacteria [29].

aPDT appears to be a promising tool in the fight against infections in diabetic patients due to the fact that their infections are typically long-lasting, which increases the chances of developing antibiotic resistance. Additionally, there is a higher risk of complications from long-term antibiotic therapy in these patients.

Curcumin is one of the natural photosensitizers. Muniz et al. investigated the use of antimicrobial photodynamic therapy (PDT) with curcumin in treating MRSA infections in Type 1 diabetic mice. A solution containing 100 μg of curcumin was activated with LED light (450 nm) at a fluency of 13.5 J/cm^3^ and intradermally injected at the infection site (PDT group). Two control groups were included: one treated with saline and the other with non-photoactivated curcumin. The PDT group showed a significant reduction in bacterial load in the lymph node compared to the saline and curcumin groups (*p* < 0.05) 24 h after treatment. Additionally, the PDT group exhibited higher levels of nitrates and nitrites (*p* < 0.001) and less intense myeloperoxidase expression (*p* < 0.001) at the infection site. Cytokine levels (IL-1β, IL-12, IL-10) were also lower in the PDT group, indicating a reduced inflammatory response. This pilot study demonstrates the therapeutic potential of intradermally administered PDT with curcumin in treating *S. aureus* infections in type 1 diabetic mice [30].

aPDT has long been a subject of interest for centers treating periodontal diseases. Cláudio et al. in their study focusing on the treatment of periodontitis in patients with uncompensated type 2 diabetes showed that adding aPDT to the standard antibiotic treatment protocol resulted in better treatment outcomes. They demonstrated a beneficial effect on reducing inflammation of periodontal tissue and lowering the risk of disease progression, as well as recurrence of the disease over several years [31].

Cunha et al. demonstrated that adjuvant aPDT in non-surgical periodontal treatment reduced clinical periodontal parameters and inflammatory cytokines in both type 1 diabetes mellitus patients and normoglycemic patients. However, normoglycemic patients with periodontitis showed a more favorable response to adjuvant aPDT treatment. They emphasized the importance of initial diabetes control for prognosis, highlighting its significant role in achieving better treatment outcomes [32].

However, two meta-analyses have shown that aPDT may not have such a strong effect on periodontal treatment. These analyses, however, were based on highly heterogeneous study groups, which made it difficult to establish reliable correlations or draw definitive conclusions [33,34].

The use of aPDT specifically for the treatment of diabetic foot ulcers seems to be a new and promising approach worth attention. Given its potential to reduce infection, promote healing, and modulate inflammation, aPDT could offer significant benefits in the management of chronic diabetic ulcers, which are also often difficult to treat with traditional methods.

Morley et al. conducted a blinded, randomized, placebo-controlled Phase IIa trial which included 16 patients with chronic leg ulcers and 16 with diabetic foot ulcers (eight per group receiving active treatment or placebo). All ulcers had persisted for over three months and were colonized with more than 10⁴ CFU/cm^2^. After assessing bacterial load, wounds were treated with either cationic photosensitizer PPA904 [3,7-bis(N,N-dibutylamino) phenothiazin-5-ium bromide] or placebo for 15 min, followed by 50 J/cm^2^ of red light, then re-sampled for bacterial analysis. Ulcer size was monitored for three months. The treatment was well tolerated, with no reported pain or safety concerns. Unlike the placebo group, actively treated patients showed a significant bacterial reduction post-treatment (*p* < 0.001). After three months, complete healing occurred in 50% of chronic leg ulcer patients receiving treatment, compared to 12% in the placebo group [35].

Carrinho et al. demonstrated benefits of using aPDT with methylene blue in the treatment of diabetic ulcers. During the experiment, all patients were treated with collagenase/chloramphenicol, while 50% of them also received PDT with methylene blue (0.01%) and laser therapy (660 nm, 30 mW, 8 sec, 6 J/cm^2^, 0.04 mm^2^ beam) three times per week for 10 sessions. Ulcer areas were measured, photographed, and analyzed using ImageJ software. Results showed a significant difference (*p* < 0.05) between the PDT and control groups, with PDT leading to greater ulcer reduction [36].

Another study, also examining aPDT with the use of methylene blue, conducted by Ferreira et al. evaluated the clinical progression of patients with diabetic foot ulcers treated with aPDT using the Bates-Jensen (BJ) scale. A total of 21 patients were monitored, with an average age of 58 years. The patients underwent the standard treatment protocol of the institution, supplemented with aPDT using 0.01% methylene blue and laser irradiation (660 nm, 100 mW, 6 J per point). The treatment protocol involved applying irradiation directly to the exposed wound surfaces at 1 cm intervals, using a laser positioned perpendicularly at a 90-degree angle. Prior to this, the treated area was infused with 0.01% methylene blue for 5 min. After the procedure, silver dressings were used to cover the wounds. The median number of aPDT sessions was eight, with the number of sessions ranging from a minimum of four to a maximum of 13. One patient (4.7%) showed health decline, which required discontinuation of the protocol. The application of methylene blue in aPDT was found to be an effective, especially in decreasing ulcer lesion area, safe, and well-tolerated treatment option, demonstrating high patient compliance and the potential to be integrated into the management of diabetic foot ulcers. Studies of this type should be repeated with a larger group of patients to enable a more reliable analysis, particularly one that includes a control group [37].

Martinelli et al. utilized a previously studied substance, RLP068—a cationic zinc phthalocyanine derivative activated by red light—in their clinical case analysis. RLP068 is known for its antibacterial and antifungal properties, as well as its low risk of resistance development. In the study, photodynamic therapy (PDT) with RLP068 promoted ulcer healing and significantly reduced the infection burden. The treatment was well tolerated, aligning with prior research findings. PDT proved to be a valuable antimicrobial option, particularly for patients undergoing multiple drug treatments, those with ulcers infected by drug-resistant bacteria, or as a complementary approach alongside other therapies [38].

5-Aminolevulinic acid (ALA) is widely utilized in dermatology and oncology as an effective component of photodynamic therapy (PDT). A clinical study conducted by Li et al. suggests that ALA may also hold promise for the treatment of diabetic ulcers. Their findings demonstrated that all patients achieved infection control and remained relapse-free during follow-up. The treatment protocol involved initial irradiation of the wounds with 20% ALA-PDT (635 nm, 100 J/cm^2^, 80 mW/cm^2^) using a red LED light source to manage infection. Additionally, if granulation necrosis or wound exudate was present, the therapy was supplemented with debridement. PDT sessions were administered weekly and continued until complete DFU healing was achieved [39].

A meta-analysis conducted by Hou et al. compared photodynamic therapy (PDT) with the standard of care (SOC) in the management of diabetic foot ulcers. The findings confirmed that PDT is an effective therapeutic modality, leading to a significant reduction in ulcer size, improved healing rates, and lower patient-reported pain levels [40].

In summary, antimicrobial photodynamic therapy (aPDT) used in the treatment of diabetic ulcers represents a promising method with potential for infection reduction and wound healing. It is a minimally invasive therapy that utilizes photosensitizers activated by light to effectively eliminate pathogens, reduce inflammation, and support regenerative processes. aPDT offers many advantages, such as selectivity in targeting bacteria, minimizing the risk of resistance, and potentially improving the quality of life for patients with chronic wounds. However, although preliminary results are promising, this method requires further research to better understand its mechanisms, optimize treatment parameters, and confirm its long-term effectiveness in treating diabetic ulcers.

### 2.2. Liposomal Photocatalytic Carriers

Currently, LNPs are being widely explored for their role in stimulating angiogenesis and delivering therapeutic agents directly to the wound site, enhancing the healing process in diabetic foot ulcers and other chronic wounds. Lipid-based nanocarriers, including liposomes, niosomes, ethosomes, solid lipid nanoparticles, and lipidoid nanoparticles, have been extensively investigated for their potential in foot ulcer therapy, as supported by relevant research studies. These nanodrug delivery systems have exhibited significant wound-healing efficacy, particularly in diabetic conditions, due to the enhanced therapeutic action of the encapsulated bioactive agents [41]. The intrinsic cavity structure of lipid nanoparticles (LNPs) provides a distinct advantage for various molecules loading, while their lipid membrane closely resembles the phospholipid bilayer of cells, contributing to excellent biocompatibility. Their nanoscale size allows them to penetrate deep into the wound microenvironment, facilitating effective transport. As a result, LNPs are widely recognized as highly efficient carriers [42]. Several studies have observed the effective action of LNP-based drug delivery systems (DDSs) in promoting angiogenesis in diabetic wounds. Due to their unique properties and flexible design, LNPs enable precise and controlled release of angiogenic factors, making them a promising approach to transforming the treatment of diabetic wounds and improving patient healing outcomes [43,44]. As the use of LNPs appears to hold growing potential in the treatment of DFUs, it is important to examine particles from this category that could be effectively utilized as carriers in the photocatalysis process. Currently, liposomes appear to be suitable for this purpose, and they have been used in previous studies for such purpose. Liposomes are spherical vesicles with a phospholipid bilayer, containing an aqueous core, made from natural or synthetic lipids and have proven effective as drug carriers in treating wounds like second-degree burns, cuts, and chronic wounds. They cover the wound, deliver the active drug, and promote healing by maintaining a moist environment. Their ability to retain moisture at the wound site is a fundamental key advantage, making them an excellent dressing base material due to their physicochemical properties [45].

Wei et al. used carbon dot liposomes (CD_somes_) to investigate their effectiveness in wound healing on mice, aiming to apply this method for the treatment of diabetic wounds. The study described the synthesis of amphiphilic carbon dots (CDs) using a one-step pyrolysis method. These CDs exhibit photoreactive properties, switching between “on” and “off” states under UV (385 nm) and green light (532 nm) irradiation. In the “on” state, exposure to UV light generates electron–hole pairs that catalyze the production of hydrogen peroxide (H_2_O_2_). In the “off” state, metastable CDsomes, acting like peroxidase, convert H_2_O_2_ into hydroxyl radicals (•OH) under green light. This light-triggered, programmable reaction allows CDsomes to participate in a cascade reaction, effectively eliminating a broad spectrum of bacteria, including methicillin-resistant *Staphylococcus aureus* (MRSA). In a study on mice with diabetic wounds infected with MRSA, this therapy effectively eradicated pathogens from the infection site and promoted angiogenesis, epithelialization, and collagen deposition. In addition to their antibacterial effects, CDsomes accelerated wound healing by modulating the immune response—reducing pro-inflammatory and increasing anti-inflammatory cytokines, which was attributed to the presence of oleic acid in the CDsomes’ structure. The study evaluated the antibacterial effectiveness of CDsomes against *E. coli*, *P. aeruginosa*, S. aureus, and MRSA by measuring MIC90 values (the minimum concentration required to inhibit 90% of bacterial growth) using the broth dilution method. Without light activation, MIC90 values were 1.7–2.5 μg/mL for Gram-positive bacteria and >1000 μg/mL for Gram-negative bacteria. UV exposure reduced MIC90 for Gram-negative strains, whereas green light had little impact. The most effective approach was sequential exposure to UV and green light, which lowered MIC90 for Gram-negative bacteria to 104.1–112.4 μg/mL. The enhanced efficacy against Gram-positive bacteria was attributed to easier cellular uptake of CDsomes, facilitated by teichoic acids in their cell walls. Meanwhile, the improved antibacterial effect against Gram-negative bacteria was linked to cascade photocatalytic reactions generating reactive oxygen species (ROS) [46].

Maintaining a balance between the production and elimination of reactive oxygen species (ROS) is critical for effective infection control and tissue regeneration. Although ROS play a pivotal role in immune response and cellular signaling, excessive ROS accumulation can impair wound healing by inducing oxidative stress and cellular damage. Inspired by the natural process of photosynthesis, Wan et al. developed and demonstrated the effectiveness of a model designed to modulate ROS levels within the wound-healing microenvironment. This approach emulates plant mechanisms that efficiently regulate ROS during photochemical energy conversion. By maintaining ROS homeostasis, this innovative strategy promotes tissue regeneration while minimizing oxidative damage. They utilized a photo-driven H_2_-releasing liposomal nanoplatform (Lip NP) composed of an upconversion nanoparticle (UCNP) linked to gold nanoparticles (AuNPs) through an ROS-sensitive connector. Hydrogen (H_2_) acts as an antioxidant by selectively reducing highly cytotoxic ROS, such as •OH and ONOO–, in diseased cells while maintaining the normal physiological functions of ROS in healthy cells, and exerts no toxicity even at high doses. This complex is enclosed within a liposomal structure, where chlorophyll a (Chla) is embedded in the lipid bilayer. The UCNP acts as a transducer by converting near-infrared (NIR) light into upconversion luminescence, enabling simultaneous imaging and localized therapy. An NIR laser can penetrate biological tissues and is transformed into green and red upconversion luminescence (UCL) by Cit-UCNP within a nanocomplex. The green UCL is utilized for Förster Resonance Energy Transfer (FRET) imaging to assess local ROS levels, while the red UCL triggers the photosynthesis of gaseous hydrogen (H_2_) to eliminate excessive ROS. AuNPs function as light-harvesting antennas to monitor the local ROS concentration for FRET-based biosensing, while Chla facilitates hydrogen gas photosynthesis to neutralize excess ROS at the affected site. In the in vitro study, the effects of three different conditions on macrophages were compared. Natural LPSs proved to be a strong activator of macrophages, inducing the overproduction of reactive oxygen species (ROS) and increased expression of proinflammatory cytokines (IL-1β and IL-6). Treatment with a bulk solution (BS) that contained the free reacting molecules, nanocomplexes, and Chla + NIR and Lip NPs + NIR significantly reduced the excess ROS and proinflammatory cytokines in LPS-stimulated macrophages. The mechanism of action of a photo-driven H_2_-releasing liposomal nanoplatform (Lip NP), composed of an upconversion nanoparticle (UCNP) linked to gold nanoparticles (AuNPs) through an ROS-sensitive connector, and the use of a near-infrared (NIR) light laser, are presented in Figure 5. However, treatment with Lip NPs + NIR showed even stronger anti-inflammatory effects and more effectively reduced the excess ROS compared to BS + NIR, which was attributed to the higher production of hydrogen (H_2_). BS + NIR led to a 28.2 ± 4.4% reduction in H_2_O_2_ levels compared to a 65.7 ± 3.7% reduction achieved by Lip NPs + NIR [47]. The application of this type of therapy in clinical studies appears to be a quite promising solution for the treatment of hard-to-heal wounds, such as diabetic foot ulcers, due to its favorable action profile, high effectiveness in maintaining ROS balance in the diabetic foot ulcers environment, and the practical non-toxicity of the substances used and the generated hydrogen.

### 2.3. Photocatalysis Against High Glucose Level Environment

Chronic hyperglycemia, characteristic of diabetes, triggers the activation of multiple molecular pathways that sustain elevated blood glucose levels. A key consequence is the excessive production of reactive oxygen species (ROS), leading to oxidative stress and cellular damage. Oxidative stress activates pathways such as the polyol and hexosamine pathways, as well as the overactivation of protein kinase C (PKC) isoforms, and promotes the formation of advanced glycation end-products (AGEs), exacerbating inflammation and insulin resistance. Hyperglycemia also induces epigenetic changes that suppress the expression of genes involved in antioxidant defense mechanisms, further amplifying oxidative stress. The accumulation of ROS leads to mitochondrial damage, impairing oxidative phosphorylation and the tricarboxylic acid (TCA) cycle, resulting in ATP deficiency. Mitochondrial dysfunction in diabetic wounds hampers key repair processes—such as cell proliferation, migration, and collagen synthesis—and sustains chronic inflammation, promoting cellular senescence and apoptosis, which significantly delays wound healing [48].

Photocatalysis can provide therapeutic benefits and improve the clinical condition of patients not only through its antibacterial effects but also by reducing local glucose concentration. This is particularly important, as glucose levels strongly correlate with the extent of ulceration and the duration of the healing process.

Hydrogen molecules have been recognized as safe and effective anti-inflammatory agents capable of alleviating ischemia-reperfusion injury and activating skin cells to promote wound healing. Hydrogen-incorporated titanium oxide nanorods (HTONs), characterized by a uniform rutile crystalline structure, function as visible-light-sensitive photocatalysts with potential therapeutic applications. The mechanism of action of HTONs is illustrated in Figure 6. The incorporation of hydrogen results in a reduction in the electronic bandgap, enhancing the material’s ability to efficiently absorb visible light, particularly in the red and near-infrared (NIR) spectra. Upon exposure to light electrons, HTONs are excited from the valence band to the conduction band, creating electron–hole pairs. HTONs exhibit potent oxidative properties, facilitating the direct oxidation of glucose molecules present in the wound or tissue microenvironment. In the reaction mechanism, HTONs accept an electron from the glucose molecule (C_6_H_12_O_6_), leading to the partial oxidation of glucose to products such as gluconic acid (C_6_H_12_O_7_) and, in consequence, other compounds in the redox reaction cascade. This process results in a local decrease in glucose concentration as glucose is consumed in the photocatalytic reaction mediated by HTONs. Moreover, HTONs generate reactive oxygen species (ROS), including hydroxyl radicals (•OH), singlet oxygen (^1^O_2_), and superoxide anions (O_2_^−^•). These ROS are highly reactive and can further oxidize glucose molecules, accelerating their degradation into less harmful products. Consequently, the effect of glucose reduction is significantly amplified. The reduction in local glucose concentrations within the microenvironment diminishes the available substrate for the formation of advanced glycation end-products (AGEs), which are typically produced via non-enzymatic glycation of proteins and lipids. Additionally, the presence of ROS may disrupt the formation of AGEs by degrading their precursors. Photocatalytic depletion of glucose, coupled with the generation of molecular hydrogen, synergistically attenuates the formation of advanced glycation end-products (AGEs) and downregulates the expression of their receptor (RAGE) within the diabetic wound microenvironment. This coordinated modulation mitigates skin cell apoptosis while promoting cellular proliferation and migration, ultimately enhancing the regenerative processes essential for effective diabetic wound healing.

Chen et al., in a study on a mice model, used photocatalytic hydrogen generation using HTONs under visible light exposure. The findings demonstrated that local glucose degradation and hydrogen generation via photocatalysis significantly accelerated diabetic wound healing, presenting a promising strategy for the treatment of diabetic foot ulcers. Furthermore, the developed HTON-hydrogel dressing is user-friendly and safe, highlighting its strong potential for future clinical studies with patients [49].

The application of photocatalysis for lowering overall glucose levels is also an area of research interest. Dafrawy et al. conducted a study on rats aimed at investigating the effectiveness of using a ZnO/poly(vinyl alcohol) (PVA) nanocomposite under ultraviolet-visible (UV–vis) irradiation in lowering blood glucose levels. The results of the experiment showed that treatment with this combination led to a significant reduction in blood glucose levels in the rats in the experimental group. ZnO/PVA, when exposed to UV–vis irradiation, demonstrated potential in modulating metabolic processes related to excessive glucose production in the experimental subjects. The application of this nanocomposite could represent a promising therapeutic strategy, especially in the context of localized therapy, such as in the treatment of DFUs. The reduction in glucose levels in the rats suggests the potential for utilizing this solution in future clinical studies, which could involve the use of ZnO/PVA in treating diabetic ulcers and other complications related to hyperglycemia, such as chronic inflammation or impaired wound healing. Therefore, therapy using ZnO/PVA nanocomposites may become a promising tool in future strategies for treating chronic wounds, particularly in the context of diabetes, where controlling blood glucose levels is crucial for the healing process [50].

## 3. Combining aPDT with Antibiotics to Obtain Better Treatment Effects

Modern medicine is continuously exploring innovative therapeutic strategies to enhance the efficacy of existing treatments. One such approach involves the integration of antibacterial photodynamic therapy (aPDT) with antibiotic therapy, which has the potential to be particularly beneficial in the management of infected diabetic foot ulcers (DFUs). Diabetic foot infections not only impair ulcer healing but also increase the risk of amputations due to their ability to spread to adjacent tissues. Additionally, these infections are often polymicrobial and exhibit high levels of antibiotic resistance, making their treatment particularly challenging.

The adjunctive use of aPDT alongside conventional antibiotic therapy may lead to improved treatment outcomes by effectively eliminating a broad spectrum of bacterial species and disrupting their metabolic processes, especially when dealing with multi-drug resistant bacteria. Furthermore, numerous in vitro studies have demonstrated that aPDT is effective against bacterial biofilms [51,52]. Since biofilms act as a major barrier to antibiotic penetration, aPDT-mediated biofilm disruption may facilitate deeper antibiotic diffusion, thereby significantly reducing the minimum inhibitory concentration (MIC) required for bacterial eradication.

Willis et al. conducted research based on the potential synergistic effect of such a combination and studied this idea on MRSA strains USA300 and RN4220. In this case, methylene blue-based aPDT was combined with ampicillin, kanamycin, tetracycline, and chloramphenicol.

The study demonstrated that antibacterial photodynamic therapy (aPDT), when combined with antibiotics, enhances bacterial eradication without inducing heritable resistance. The primary mechanism involves membrane disruption and reactive oxygen species (ROS) generation, leading to a temporary reduction in bacterial resistance. Additionally, Willis and collaborators observed that aPDT disrupts bacterial efflux pump activity, impairing antibiotic extrusion mechanisms. This localized membrane damage enhances intracellular antibiotic retention, further reducing the required antibiotic dosage for effective bacterial killing. FICI values indicated an additive effect across all tested combinations, suggesting potential for broader ARB treatment. Safety analysis indicated that aPDT can lower bacterial resistance to susceptible levels, supporting its clinical feasibility in localized infections.

A short analysis was conducted to evaluate the clinical safety of combined aPDT, considering the required photosensitizer concentration and total light exposure. For example, tetracycline treatment for the USA300 strain required an average exposure of 5.21 J/cm^2^ to reduce resistance to susceptible levels. Since this dose is lower than typical clinical applications, the combination of aPDT and antibiotics appears to be a safe and effective method for preserving the efficacy of current antibiotics.

The broadening of MIC distribution with increasing aPDT doses improves bacterial elimination, making even moderate aPDT doses a promising strategy to support antibiotic therapies and combat antibiotic-resistant bacterial strains [53]. The additive effect of antibiotics and aPDT is shown in Figure 7.

According to a meta-analysis conducted in 2021, the most frequently isolated microorganisms in DFU infections include *Staphylococcus aureus* (with methicillin-resistant strains (MRSA) accounting for 18.0%), *Pseudomonas* spp., *Escherichia coli*, and *Enterococcus* spp. [54]. Some of those pathogens have been included in numerous studies evaluating the combination of aPDT and various antibiotics.

### 3.1. Gentamicin

In a 2021 study, Nieves et al. explored the combination of gentamicin with a novel photoantimicrobial agent. The conjugate was created through a click reaction between gentamicin and the red-light absorbing 9-isothiocyanate-2,7,12,17-tetrakis(methoxyethyl)porphycene (9-ITMPo), resulting in the formation of ATAZTMPo-gentamicin, a new antimicrobial agent for antimicrobial photodynamic therapy (aPDT). The conjugate’s efficacy was tested against Gram-positive (*Staphylococcus aureus*) and Gram-negative (*Escherichia coli*) bacterial strains. In vitro results showed that ATAZTMPo-gentamicin is a powerful broad-spectrum near-infrared (near-IR) photoantimicrobial agent, displaying no dark toxicity and remaining effective at submicromolar concentrations. The bactericidal activity is attributed to the photodynamic action of the porphycene, while gentamicin improves its solubility and amphiphilicity, aiding in the conjugate’s ability to penetrate and disrupt both the bacterial outer membrane and internal structures, enhancing its antimicrobial effect. These findings suggest that ATAZTMPo-gentamicin enables the use of lower doses than gentamicin alone while still achieving bactericidal effects [55].

A study by Barra et al. explored the use of a combination of antimicrobial photodynamic therapy (aPDT) and gentamicin for treating antibiotic-resistant and biofilm-forming bacterial strains. The research focused on *Staphylococcus aureus*, *Staphylococcus epidermidis*, and *Staphylococcus haemolyticus* biofilm strains. The findings revealed that gentamicin alone was ineffective against these strains, as evidenced by very high MIC90 values. In the aPDT portion of the treatment, 5-aminolevulinic acid (5-ALA), a pro-drug, was used. Once absorbed by proliferating bacteria, 5-ALA is converted into the natural photosensitizer protoporphyrin IX (PpIX). The effects of the treatment were examined through confocal microscopy. Imaging of biofilms after 5-ALA/PDT treatment showed that bacteria metabolized 5-ALA, resulting in the distribution of PpIX within the cells. While reactive oxygen species (ROS) generated by aPDT killed the majority of the bacteria, some survived with compromised membranes and enzymes. This damage enhanced the penetration of gentamicin, thereby improving its efficacy when applied after the aPDT treatment [56].

### 3.2. Ciprofloxacin

Ronqui et al. investigated the potential synergistic effects of antibacterial photodynamic therapy (aPDT) combined with ciprofloxacin. The study aimed to assess the effects of aPDT using methylene blue (MB) on *Staphylococcus aureus* (ATCC 25923) and *Escherichia coli* (ATCC 25922) in both biofilm and planktonic states.

The results of the study demonstrated that the combination of aPDT and ciprofloxacin offers several advantages, especially when aPDT is administered prior to antibiotic treatment. In this context, the use of sub-inhibitory concentrations (below the minimum inhibitory concentration, MIC) of ciprofloxacin, along with lower doses of MB and reduced light fluencies, achieved the highest bacterial reduction. The sequential application of aPDT before ciprofloxacin likely caused sufficient damage to the bacterial cell wall, thereby enhancing the uptake of the antibiotic and augmenting the overall antibacterial effect.

However, bacteria in the biofilm state showed resistance to aPDT using MB. In biofilm assays, no significant bacterial reduction was observed with MB treatment at a maximum concentration of 400 μg/mL or with light exposure at a maximum of 22.4 J/cm^2^. Despite the greater resistance of biofilm-associated bacteria to aPDT compared to planktonic forms, aPDT still induced morphological changes in the biofilm structure, which enhanced the antibacterial effects of ciprofloxacin. This finding underscores that the combination of both therapies is more effective than using either modality as monotherapy. Additionally, the study observed that Gram-positive bacteria were more susceptible to aPDT than Gram-negative species [57].

### 3.3. Imipenem

Feng et al. identified antimicrobial photodynamic therapy (aPDT) as a potential strategy to address one of the key mechanisms of bacterial antibiotic resistance: the production of carbapenemases. In this context, aPDT could significantly enhance the efficacy of carbapenems against bacterial strains that primarily produce carbapenemases, thereby neutralizing their resistance mechanisms and enabling more effective therapeutic applications.

Methylene blue was selected as the photosensitizer for aPDT in the study. The bacterial strains tested included *Staphylococcus aureus* without carbapenemase production, as well as three bacterial strains producing different types of carbapenemases-class A, B, and D. Imipenem, as a representative of the carbapenem class of antibiotics, was used to evaluate potential effects.

The results demonstrated that aPDT effectively inactivated bacterial carbapenemases and reduced bacterial viability in carbapenemase-producing strains, thereby increasing the effectiveness of imipenem against them. This included strains harboring class A, B, and D carbapenemases. The treatment also damaged the genetic determinants responsible for carbapenemase production, potentially preventing their transmission. Furthermore, the additional effects of aPDT on bacterial strains enhanced antibiotic efficacy by increasing their sensitivity, even at sub-lethal doses [58].

### 3.4. Vancomycin

Mills et al. proposed a conjugate specific for Gram-positive bacteria. An antimicrobial photodynamic therapy agent specific for this bacterial group (VanB2) was created by combining a photosensitizer for aPDT ryboflavin with antibiotic vancomycin.

Results of the study showed that VanB2 displayed potent photodynamic antibacterial activity, effective against Gram-positive pathogens, including vancomycin-resistant Enterococcus (VRE) and MRSA, even at submicromolar concentrations. It also overcame resistance mechanisms and demonstrated efficacy against bacterial biofilms. The conjugate enhances vancomycin’s activity by ~10 times, making it a promising tool for photodynamic therapy and light-activated prodrug strategies.

Additionally, VanB2 demonstrated a good safety profile, showing no hemolytic activity in human erythrocytes and low toxicity to HaCaT keratinocytes at concentrations effective for bacterial killing [59].

### 3.5. Ceftriaxone

Magacho et al., in 2020 during an in vitro study, proposed yet another combination of aPDT with antibiotics in the form of methylene blue associated with ceftriaxone. The study focused on Gram-negative bacteria—*Klebsiella pneumoniae*, *Enterobacter aerogenes*, and *Escherichia coli*.

In the study, the results were not as promising as the results previously mentioned. PDT-MB alone and in combination with Ceftriaxone produced similar effects in reducing bacterial growth. Thus, PDT-MB shows potential as an alternative for inactivating Gram-negative strains, although the combination with ceftriaxone did not enhance its efficacy [60].

The studies discussed above demonstrate that combining novel antimicrobial photodynamic therapies (aPDTs) with antibiotic treatment can enhance the eradication of both Gram-positive and Gram-negative bacteria. Certain combinations of these therapies show greater potential than others. To facilitate the potential implementation of such treatment strategies for diabetic foot ulcer (DFU) infections, further studies on the combination of aPDT with antibiotics recommended by the International Working Group on the Diabetic Foot (IWGDF) and the Infectious Diseases Society of America (IDSA) are essential. Comprehensive research on the application of these combinations in treating DFUs, in light of the growing challenge of multi-drug-resistant bacteria and the increasing prevalence of diabetes mellitus, could lead to highly beneficial treatment options for managing DFU infections.

## 4. Other Potential Combination Therapies with Photodynamic Therapy to Enhance Treatment and Receive Better Healing Outcomes

In recent years, bacterial drug resistance has become a major problem in many fields of medicine [61]. This issue significantly affects the treatment of diabetic ulcers, as the number of DFU infections caused by multi-drug-resistant bacteria is constantly rising [62].

To address this growing concern, various combination treatments, including aPDT, have been proposed to enhance current DFU infection therapies.

### 4.1. Antimicrobial Photodynamic Therapy Combined with Silver Nanoparticles

A 2021 study by Akhtar et al. investigated the effectiveness of photodynamic therapy mediated by novel toluidine blue-conjugated, chitosan-coated gold–silver core–shell nanoparticles (TBO–chit–Au–AgNPs) to evaluate its potential for treating diabetic ulcers.

In recent years, silver nanoparticles have been proposed as a promising option for the treatment of diabetic foot ulcers (DFUs). Several recent reviews suggest that silver nanoparticles demonstrate potential in improving DFU management, outperforming conventional dressings in terms of infection control, healing time, and frequency of dressing changes. However, they have not been shown to provide superior outcomes in limb salvage. Despite the lack of sufficient clinical evidence for their overall effectiveness in DFU healing, some studies indicate that silver nanoparticles may positively influence wound-healing rates, particularly when combined with other therapeutic modalities [63,64].

The effectiveness of this combination with aPDT was tested in vitro on *Staphylococcus aureus* and *Pseudomonas aeruginosa* biofilms (both monomicrobial and polymicrobial). Additionally, its efficacy was evaluated in an in vivo study on rats with induced diabetic foot ulcers infected with the same bacterial strains used in vitro. Results proved the potential of this therapy as the TBO–chit–Au–AgNPs-mediated photodynamic therapy effectively eliminated multi-drug-resistant Gram-positive and Gram-negative biofilms (monomicrobial and polymicrobial). This innovative nano-phototheranostic complex was also shown to be a nontoxic antibacterial agent [65].

Parasuraman et al. conducted research about the application of silver nanoparticles (AgNPs) in order to amplify aPDT against *Staphylococcus aureus* and *Pseudomonas aeruginosa*. The study revealed that silver nanoparticles coated with methylene blue (MB-AgNPs) demonstrated superior antibacterial and anti-biofilm properties compared to both free MB and AgNPs. When exposed to light, MB-AgNPs significantly reduced the viable bacterial counts of S. aureus and *P. aeruginosa*. The interaction between MB and AgNPs led to an increase in reactive oxygen species (ROS) production, thereby enhancing phototoxicity and therapeutic efficacy.

Additionally, MB-AgNPs were able to penetrate deeper into bacterial biofilms, enhancing their biofilm-eradicating activity compared to free MB. Photodynamic therapy utilizing these nanoparticles proved more effective in biofilm reduction, indicating their potential as a promising alternative for treating antibiotic-resistant bacterial infections [66].

When considering the application of silver nanoparticles (AgNPs) in biomedical contexts, it is crucial to thoroughly assess their long-term biosafety, particularly with respect to their impact on metabolic pathways in human systems. Although numerous studies highlight their beneficial properties, such as potent antimicrobial activity, growing evidence also points to their potential cytotoxicity and metabolic disruption.

In vitro investigations, as summarized by Noga et al., emphasize that AgNP-induced toxicity is primarily associated with the generation of reactive oxygen species (ROS), leading to oxidative stress, DNA damage, and activation of apoptotic pathways. Exposure to AgNPs has been shown to impair mitochondrial function, alter energy metabolism (reduced ATP production), and cause DNA fragmentation and membrane damage across various cell types, with smaller nanoparticles exhibiting higher cytotoxic potential [67].

In vivo studies have further confirmed these findings. Haiyun et al. demonstrated that Au@Ag nanoparticles not only induced mitochondrial and lysosomal dysfunction, but also significantly disrupted metabolic processes in the liver, as evidenced by the alteration of 27 metabolites involved in lipid, amino acid, and choline metabolism. These findings suggest a direct interference of AgNPs with critical biochemical pathways, ultimately impairing hepatic metabolic homeostasis and promoting inflammatory responses through activation of the NLRP3 inflammasome [68].

Al-Doaiss et al. similarly conducted research on mice exposed to 10 nm AgNPs for 35 days, revealing pronounced hepatic dysfunction, including Kupffer cell activation, hepatocyte necrosis, glycogen depletion, vascular injury, and oxidative stress-mediated inflammation. The observed glycogen depletion and hepatocellular damage point to impaired carbohydrate metabolism and liver energy balance [69].

Another study by Nosrati et al., involving rats treated with AgNPs over 28 days, reported histopathological evidence of renal damage, including glomerular degeneration, tubular injury, and fibrosis, alongside the deregulation of key genes involved in cell survival, apoptosis, and inflammation (Bcl-2, Bax, EGF, TNF-α, TGF-β1). These molecular alterations further implicate AgNPs in the disruption of normal renal metabolic pathways [70].

Other studies have also demonstrated AgNP-induced toxicity in organs such as the lungs and reproductive system, often linked to metabolic dysregulation and oxidative stress. In summary, while silver nanoparticles show significant biomedical promise, careful consideration of their potential to disrupt vital metabolic pathways is essential. Further in vivo investigations focusing on metabolic processes and long-term organ function, particularly in the liver and kidneys, are crucial for the safe clinical application of AgNPs, especially when combined with aPDT.

### 4.2. Antimicrobial Photodynamic Therapy Combined with Wound Dressings

A study from 2024 concluded by Brandão et al. proposed a treatment protocol consisting of an antimicrobial photodynamict therapy (aPDT) with a 20% sodium chloride dressing (Mesalt^®^). Brandão et al. highlited the problem of hypergranulation in the healing process of diabetic foot ulcers.

Hypergranulation occurs when immature granulation tissue grows excessively beyond the surrounding skin, preventing proper re-epithelialization. This condition is commonly associated with various risk factors, such as wound healing by secondary intention, chronic inflammation due to high bacterial load, hypoxia from occlusive dressings, excessive wound drainage, and friction. These factors disrupt the normal healing process and can lead to delayed recovery [71].

In order to deal with this problem, the newly proposed protocol was implemented in the treatment of a 62-year-old female from São Paulo, Brazil. She was diagnosed with diabetes 34 years ago and has been using insulin therapy. Her A1c was 7.3%, and the circulation in both lower limbs was normal. A diabetic foot ulcer (DFU) developed in August 2023 on her left hallux after a previous amputation at the same site. Inadequate care and footwear likely led to the new ulcer. The DFU was circular and treated with collagenase, with family members performing dressing changes at home. The left foot was dry, and both feet had positive protective plantar sensitivity. Microbiological analysis detected *Staphylococcus aureus*. The protocol involved daily wound cleansing with saline, applying wet gauze with 20% sodium chloride for five minutes, followed by Mesalt^®^ as a secondary dressing. Weekly follow-ups at the wound care clinic included aPDT and reapplication of Mesalt^®^. Results of this form of treatment were significant as the wound-healing process considerably improved. Hypergranulation decreased and the diabetic ulcer showed lower exudate levels which proved this combined therapy’s potential [72].

### 4.3. Antimicrobial Photocatalytic Therapy Combined with Anti-Hipoxia Mechanism

One of the challenges encountered in patients with diabetic ulcers is the inadequate supply of oxygen required for the proper regeneration of healing tissues. To optimize treatment and resolve this critical aspect of wound healing, Li et al. investigated the combination of photo-eradication of pathogens with therapy targeting local hypoxia in their study. In the mouse model experiment, they used octahedral Rh/Ag_2_MoO_4_ and Rh/Ag_2_MoO_4_, along with 808 nm laser exposure, to enhance wound healing. They observed the eradication of Methicillin-resistant *Staphylococcus aureus* (MRSA) and *E. coli* due to Rh/Ag_2_MoO_4_ photothermal and photocatalytic antibacterial action, in comparison to the control groups. Moreover, on days 4 and 10 of the study, they measured indicators important for promoting proper tissue regeneration. They found a reduction in HIF-1*α* expression, which is a marker of hypoxia progression, as well as an increase in VEGF expression, which suggests enhanced vascular remodeling. Rh, as a noble metal, possesses enzyme-like properties (e.g., peroxidase (POD), catalase (CAT), superoxide dismutase) and exhibits distinct photophysical properties, such as photothermal and photocatalytic antibacterial activity. For this reason, the octagonal Rh/Ag_2_MoO_4_, stabilized by the presence of Rh, can effectively mitigate the hypoxic infected microenvironment via CAT-like activity and POD-like activity, in addition to its antibacterial properties. In infected wounds, excessive production of H_2_O_2_ occurs through various mechanisms, such as respiratory bursts of white blood cells. The acidic environment in such wounds facilitates the catalytic conversion of H_2_O_2_ with the involvement of Rh/Ag_2_MoO_4_, which causes significant hypoxia relief and promotes tissue regeneration by regulating the proper formation of collagen and the process of angiogenesis. The findings of their study indicated that Rh/Ag_2_MoO_4_ demonstrates strong antibacterial and anti-biofilm activity properties and significantly enhances the healing of chronic wounds, encouraging neovascularization [73].

The study conducted by Su et al. explored the use of a photocatalytic oxygen-releasing and antibacterial membrane (PMP@GFs) for the treatment of diabetic wounds. The membrane demonstrated strong antibacterial activity, effectively killing *Staphylococcus aureus* and *Escherichia coli* with killing efficiencies of 95% and 98%, respectively, while also showing significant anti-biofilm activity. Crucially, their in vitro results also showed that because of the oxygen evolution, the therapy promoted collagen production, VEGF realising stimulation, and enhanced HIF1-α expression, which helped improve tissue regeneration by alleviating hypoxia and stimulating angiogenesis in the affected tissues as in the study by Li et al. [11].

### 4.4. Antimicrobial Photodynamic Therapy Combined with Lysozyme

Lysozyme is a natural antibacterial protein that breaks down bacterial cell walls by targeting peptidoglycan, making it particularly effective against Gram-positive bacteria. It also contributes to antiviral defense, tissue repair, and wound healing. Different types exist, including C, I, and G variants [74].

In recent years, lysozyme has been explored as a potential tool for combating antibiotic-resistant bacteria in various medical fields. Numerous modifications and combinations have been proposed to enhance its effectiveness, particularly against Gram-negative bacteria [75].

One such application involves improving the therapeutic effects of antimicrobial photodynamic therapy (aPDT). Okamoto et al. evaluated a novel lysozyme-based photosensitizer conjugate—Lys-gold nanoclusters (Au NCs)/Rose Bengal (Lys-Au NCs/RB)—to assess its potential in this approach. As mentioned in the study, Au NC photosensitizers have already been applied for aPDT in the past but their potential effects on bacterial biofilm were still unproven. Okamoto et al. introduced a hybrid system of gold nanocluster (Au NC) photosensitizers shielded by lysozyme (Lys) and incorporating rose bengal (RB) as a photosensitive dye. Such a hybrid was intended to have enhanced ^1^O_2_ generation abilities due to resonance energy transfer (RET) in the Au NC/RB conjugate, while the addition of lysozyme was expected to enhance its antibacterial characteristics. Its potential effects were tested in vitro on S. mutans. Results proved its potential, as Lys-Au NCs/RB at concentrations of 0.1 μg/mL or greater was effective against bacteria. Those concentrations were much lower compared to the required concentrations for Au NCs alone or with other reported Au NC-based conjugates mentioned in other studies. Besides S. mutans, this novel hybrid showed effectiveness in battling some other common oral bacterial species featuring *E. coli*, A. naeslundii, P. gingivalis, and P. intermedia, showing it is effective against Gram-negative and Gram-positive bacterial species. What is more, it also compromised S. mutans biofilm formation. Furthermore, when tested against NIH3T3 fibroblasts, it showed low cytotoxicity towards such cells [76].

Li et al. proposed an alternative application for this combination. Building on their previous work, they designed an intelligent bio-inorganic nanohybrid to enhance the synergistic effects between aPDT and lysozyme. This nanohybrid, which incorporates a silica and dendritic mesoporous silica coating on upconversion nanoparticles (UCNPs), efficiently loaded methylene blue (MB) as a photosensitizer and lysozyme (LYZ). A bacterial hyaluronidase (HAase)-responsive valve, constructed via layer-by-layer (LBL) assembly of hyaluronic acid (HA) and poly-L-lysine (PLL), enabled the controlled release of LYZ. After assessing the technical aspects of LYZ release, this novel hybrid was tested against methicillin-resistant *Staphylococcus aureus* (MRSA) and multi-drug-resistant *Escherichia coli* (MDR *E. coli*), with a murine model employed for in vivo evaluation.

The results demonstrated that this nanohybrid exhibited potent antibacterial activity, achieving a >5 log10 reduction in MRSA viability in vitro. In the murine model, it effectively treated deep-tissue (5 mm thick) MRSA infections without causing any adverse effects. The treatment was most effective under near-infrared (NIR) light irradiation. This combination showed superior efficacy compared to single aPDT treatment. However, when comparing its effects on MRSA and MDR *E. coli*, the combination was more effective against MRSA, highlighting its enhanced activity against Gram-positive bacteria [77].

Further studies are needed to evaluate the potential of lysozyme and aPDT, particularly in terms of clinical safety and efficacy in wound healing. Additionally, testing on bacterial species commonly associated with diabetic foot ulcer (DFU) infections, as those demonstrated in the second study, will be crucial. With continued research, this therapeutic approach may become one of many valuable treatment options.

### 4.5. Antimicrobial Photodynamic Therapy Combined with Phage Enzymes

As another addition to aPDT when targeting resistant bacteria, in some studies, phage enzymes have been evaluated.

Bispo et al. combined antimicrobial photodynamic therapy (aPDT) with phage therapy to target methicillin-resistant *Staphylococcus aureus* (MRSA). They conjugated a Staphylococcus-specific cell-binding domain (CBD3) with a photoactivatable silicon phthalocyanine (IRDye 700DX), forming the CBD3-700DX conjugate. This conjugate successfully detected and bound to MRSA and S. epidermidis, with stronger affinity for MRSA.

Photoactivation experiments showed that CBD3-700DX at 0.64 mM and 2.6 mM completely eradicated both bacteria. It also disrupted biofilms by eliminating surface bacteria and reducing overall density, though deeper layers retained some viable cells. Tests in HeLa cells revealed phototoxic effects, especially in MRSA-infected cells, which may be beneficial in wound infections like diabetic foot ulcers (DFUs) by exposing intracellular bacteria [78].

Petrosino et al. developed a modular phage vector platform combining phage therapy and antimicrobial photodynamic therapy (aPDT) for targeted Gram-negative bacterial eradication. They functionalized the M13 bacteriophage capsid with Rose Bengal (RB) photosensitizers and directed it toward specific bacteria by displaying targeting peptides on the pIII coat protein.

Initial tests with a wild-type M13-RB conjugate successfully eradicated *Escherichia coli* but showed limited effectiveness against *Acinetobacter baumannii* and *Pseudomonas aeruginosa*. Retargeting the phage improved its activity against *A. baumannii*, eradicating 75% of cells at 0.25 mM RB after photostimulation. A broader retargeted version, M13Gram-RB, bound effectively to *A. baumannii* (100%) and *P. aeruginosa* (70%) but minimally to *Staphylococcus aureus* (7%), confirming selectivity for Gram-negative bacteria. Photostimulation reduced *A. baumannii* and *P. aeruginosa* survival, with the M13Aba-RB variant performing best against *A. baumannii* [79].

Both studies demonstrate significant potential in combining these two novel approaches to combat multi-drug-resistant bacterial species. Although some potential negative effects on human cells were observed in the first study, this drawback could be leveraged as an advantage in certain cases. With further studies that thoroughly evaluate the safety and potential of this approach, it may prove beneficial in the future for treating DFU infections.

### 4.6. Antimicrobial Photodynamic Therapy Combined with Potassium Iodide

A study by Bispo et al. conducted in 2021 highlighted a potential limitation of using aPDT in treatment. The research focused on aPDT with the *S. aureus-specific* immunoconjugate 1D9-700DX, which has been reported to effectively eradicate MRSA. Human plasma is responsible for natural antioxidant mechanisms in our bodies [80]. Such mechanisms disrupt the potential effects of applied aPDT therapy as this therapy relies on generating reactive oxygen species (ROS), for example, singlet oxygen (^1^O_2_). Reactive oxygen species are neutralized by those natural antixoidant characteristics of human plasma such as the presence of human serum albumin (HAS). It was proposed that this problem may be dealt with by applying potassium iodide (KI). According to Hamblin et al., potassium iodide (KI) reacts with singlet oxygen (^1^O_2_), leading to the formation of reactive iodine species, mainly molecular iodine (I_2_) and triiodide anions (I_3_^−^) [81].

In the present study, singlet oxygen (^1^O_2_) generated upon red light activation of IRDye 700DX reacts with exogenously added KI to produce molecular iodine species (I_2_/I_3_^−^). These iodine species exhibit strong antimicrobial activity independently of ^1^O_2_, allowing bacterial killing to proceed even when singlet oxygen is rapidly scavenged by antioxidants such as human serum albumin (HSA) present in plasma.

Thus, the reaction between KI and ^1^O_2_ diverts the antimicrobial effect from being dependent solely on ^1^O_2_, which is vulnerable to antioxidant neutralization, to the formation of cytotoxic iodine species that are less susceptible to antioxidant quenching, thereby enhancing aPDT efficacy in human plasma.

The results indicate that KI can counteract the antioxidant activity of human plasma during aPDT with 1D9-700DX, enabling the use of lower immunoconjugate doses and shorter irradiation times. What is more, Bispo et al. analyzed if this potential treatment did not affect mammalian cells in a negative way. Those tests performed on the human cervical cancer HeLa cell line showed that such an approach is in fact non-toxic towards them. The combination of KI and 1D9-700DX effectively targets and eliminates specific bacteria while sparing surrounding cells and non-targeted bacteria. This strategy holds promise as a complementary approach to existing MRSA treatments, with potential applications for the topical treatment of skin infections (such as DFUs) and implant-related infections [82].

Another study by Wei et al. explored the use of potassium iodide (KI) as an enhancer in antimicrobial photodynamic therapy (aPDT). The study focused on aPDT using Rose Bengal diacetate (RBDA) as a photosensitizer, activated by green light irradiation, with the addition of KI to induce enhancement mechanisms similar to those described in the previous study.

Rose Bengal aPDT has proven its potential in numerous studies. Naranjo et al. demonstrated its potential as an adjunct therapy for severe, progressive infectious keratitis prior to therapeutic keratoplasty [83]. Its safety was evaluated in rabbits by Martinez et al., showing no adverse effects compared to the control group [84].

In the study by Wei et al., this combination approach was proposed to enhance the treatment of MRSA-induced diabetic ulcer infections. Previous research has demonstrated that Rose Bengal exhibits higher efficacy against Gram-positive bacteria than Gram-negative species [85]. To evaluate the potential of KI in augmenting RBDA-mediated aPDT, Wei et al. conducted tests against MRSA, *E. coli*, and *Candida albicans*. The results demonstrated that KI significantly improved the bactericidal and fungicidal effects of RBDA, particularly against *E. coli* and *C. albicans*, which were otherwise poorly affected by RBDA alone. The addition of KI also further enhanced the already strong efficacy of RBDA against Gram-positive MRSA. The study further evaluated this approach in a diabetic mouse model with MRSA-infected wounds, showing that RBDA accelerated both bacterial clearance and wound healing, with KI providing additional therapeutic benefits [86].

If additional studies further explore the incorporation of potassium iodide (KI) to enhance the already successful antimicrobial photodynamic therapy (aPDT), this approach could significantly improve the treatment of diabetic foot ulcers (DFUs). By potentiating the efficacy of aPDT, KI may contribute to more effective bacterial eradication and enhanced wound healing, offering a promising therapeutic strategy for managing DFU infections.

## 5. Photocatalysis and Photodynamic Therapy as an Opportunity for Effective Biofilm Eradication in Diabetic Foot Ulcers—A Novel Approach

### 5.1. Biofilm Formation in Diabetic Wounds

A biofilm is a highly organized, multispecies consortium of microorganisms that confers numerous advantages, including increased resistance to adverse environmental conditions such as desiccation and nutrient depletion. From a clinical perspective, its structural complexity, combined with the presence of metabolically dormant persister cells, significantly enhances its tolerance to antimicrobial agents. Biofilm formation is initiated by the adhesion of a limited number of microbial cells to a surface, followed by extensive proliferation and the secretion of an extracellular polymeric matrix. This matrix undergoes progressive maturation, leading to the development of intricate microcolony networks with a sophisticated three-dimensional architecture, often incorporating aqueous channels, until the dispersion of planktonic cells occurs [87]. Biofilms have a crucial role in diabetic patients and contribute to delayed healing. The formation of biofilms in diabetic ulcers is influenced by several microbial and host factors. High bacterial diversity, including opportunistic and anaerobic pathogens, facilitates biofilm development, particularly in deeper wound layers where these microorganisms find a favorable niche. An increased presence of *S. aureus*, especially in neuropathic diabetic ulcers, further promotes biofilm persistence. Environmental factors such as hygiene, glycemic control, and prior antimicrobial exposure shape microbial composition, while immune dysfunction can enhance the pathogenicity of normally low-virulence bacteria. Prolonged duration and local hypoxia create conditions that favor anaerobic bacterial growth, reinforcing biofilm resilience and making infections more difficult to eradicate. Additionally, each foot ulcer develops a distinct microbiota, further complicating treatment and contributing to chronic biofilm-associated infections [88]. The conditions within chronic wounds, including the presence of necrotic tissue and debris, facilitate bacterial adhesion and biofilm formation. Unlike planktonic bacteria, biofilms are highly resistant to treatment and eradication, making them a major factor in the failure of conventional antimicrobial therapies. Biofilms are present in 60–80% of chronic wounds, significantly contributing to the persistence of infection, and are more frequently linked to Gram-negative bacteria than to Gram-positive bacteria [89].

### 5.2. Biofilm Eradication—The Use of Photocatalysis and aPDT

One of the key benefits of photocatalysis is its ability to penetrate deep into the biofilm, reaching regions that are often inaccessible to traditional antimicrobial agents. Biofilms are notoriously difficult to treat due to their complex structure, which consists of layers of microbial cells embedded in an extracellular matrix made of proteins, polysaccharides, and extracellular DNA. This matrix acts as a protective barrier, limiting the penetration of antibiotics and other therapeutic agents. In many chronic infections, particularly those associated with diabetic foot ulcers, the biofilm can be several layers deep, making conventional treatments ineffective at reaching the microorganisms within. When photocatalysts, such as titanium dioxide, are exposed to light, they generate reactive oxygen species such as hydroxyl radicals and superoxide ions, which can diffuse into the biofilm matrix and break down its components. This leads to a disruption of the protective structure of the biofilm, allowing the ROS to more effectively penetrate deeper layers and target the bacteria within. As a result, photocatalysis not only inactivates bacteria at the surface of the biofilm, but also reaches the microbial populations embedded in the more challenging deeper layers, which are often shielded from antibiotics. In chronic infections, biofilms contribute to antimicrobial resistance, as the bacteria within the biofilm exhibit reduced metabolic activity and altered gene expression, making them less susceptible to antibiotics. Additionally, the biofilm extracellular matrix can sequester antimicrobial agents, reducing their effectiveness. Photocatalysis bypasses these mechanisms, targeting the biofilm directly, and can potentially overcome the challenges posed by antimicrobial resistance. By breaking down the biofilm structure and killing or inactivating the bacteria within, photocatalysis can help restore the effectiveness of treatment, offering a promising approach for managing chronic infections that are otherwise difficult to eradicate. Furthermore, photocatalysis is used in conjunction with traditional antimicrobial treatments, enhancing their effectiveness and reducing the likelihood of resistance development [90,91]. Antimicrobial photodynamic therapy (aPDT) is a promising approach for disrupting biofilms by utilizing a synergistic combination of a photosensitizer (PS), molecular oxygen, and visible light. When exposed to light of a specific wavelength, the PS generates highly reactive oxygen species (ROS), which oxidize essential cellular components, including proteins, lipids, and nucleic acids within the biofilm matrix. This oxidative stress effectively inhibits microbial cells, even those embedded within the protective extracellular polymeric substance (EPS). aPDT selectively targets microbial cells as the PS preferentially binds to them without harming host tissues, making it a safe, non-toxic, and minimally invasive therapy. By attacking multiple components of the biofilm, this approach reduces microbial burden and biofilm formation, offering a realistic and effective strategy for managing chronic infections. Recent advancements in PS design have enhanced ROS production, further improving biofilm disruption [92,93].

### 5.3. Practical Modern Approaches to Biofilm Destruction in the Treatment of Diabetic Foot Ulcers

Biofilm eradication has become a key aspect of infection treatment in medicine in recent years. The previously mentioned treatment methods included the use of photocatalysis in therapies that also interact with biofilms. However, these approaches were within the framework of traditional methods focused on eliminating the pathogen. According to current knowledge, biofilm as a component of infection represents one of the greatest challenges in modern medicine. Therefore, research into treatments targeting biofilm is particularly important, especially in the context of the growing number of drug-resistant pathogens and the increasingly organized biofilm structures of these microorganisms, which make standard treatment approaches potentially ineffective and necessitate the use of additional techniques aimed directly at preventing and destroying biofilm structure.

One of the crucial elements seems to be the precise selection of therapy that targets the specific area without exposing unaffected tissues to oxidative stress caused by the therapy, while maintaining high effectiveness. Nanomotors, already used in medicine, demonstrate excellent selectivity and biofilm penetration, making them an ideal solution in the fight against biofilms [94]. Deng et al., in their study, explore a novel nanomotor (CSIL) that uses near-infrared (NIR) light for photothermal and photodynamic therapy to combat MRSA biofilm in diabetic wounds. The nanomotor is composed of carbon yolk with an eccentric structure and spinous shell and combines lysostaphin (an enzyme targeting MRSA) and indocyanine green, effectively eradicating MRSA biofilms and promoting wound healing. Its unique cascade photodynamic therapy strategy minimizes side effects while efficiently targeting biofilm and enhancing healing, offering a promising alternative to traditional antibiotics for treating diabetic wounds. The CSIL nanomotor helps in the healing of MRSA-infected wounds also by disrupting bacterial quorum sensing and encouraging the shift of macrophages from a pro-inflammatory (M1) to an anti-inflammatory (M2) state. Their in vitro experiments on Methicillin-resistant *Staphylococcus aureus* (MRSA) biofilms showed that CSIL combined with near-infrared (NIR) light effectively reduced bacterial colonies. Without NIR, CSIL demonstrated about 70% bactericidal activity due to lysostaphin’s specific targeting of MRSA. When NIR was applied, bacterial colonies decreased by 99.7%, enhanced by the generation of singlet oxygen from indocyanine green (ICG). Crystal violet staining further confirmed that CSIL + NIR treatment largely eradicated biofilm structures, highlighting CSIL’s strong biofilm elimination and antibacterial potential. The study expanded to include diabetic wound models infected with MRSA in mice. After creating a wound and inoculating it with MRSA, biofilm formation was observed after 48 h. Wounds were treated with various methods, including CSIL + NIR. The CSIL + NIR treatment showed superior results in clearing MRSA biofilms, improving bacterial clearance, promoting wound healing, reducing inflammation, and encouraging collagen deposition and angiogenesis. CSIL + NIR also promoted macrophage shift from M1 to M2, further enhancing tissue repair [95].

Another promising approach for effective therapy appears to be the use of immunomodulatory microneedles to target one of the key defense mechanisms of biofilms against destruction—extracellular polymeric substances (EPSs). Bacteria in biofilms are protected by EPSs, creating a barrier against antibiotics and immune cells, and exhibit specific microenvironments, such as low pH and high glutathione (GSH) levels. The ROS generated by photosensitizers can be neutralized by GSH in biofilms and limit the treatment [96]. Yang et al. propose enhancing photodynamic therapy for diabetic wounds by reducing endogenous glutathione (GSH) using SeC@PA with microneedles (MNs) for targeted delivery. Hybrid dopamine-coated nanoparticles (SeC@PA), containing selenium (Se) and chlorin e6 (Ce6), were synthesized and surface-modified with L-arginine (LA), then incorporated into a microneedle (MN) patch, forming SeC@PA MN. To assess anti-biofilm activity, *Staphylococcus aureus* (SA) and *Pseudomonas aeruginosa* (PA) were selected due to their prevalence in chronic wounds. SeC@PA(+) exhibited the strongest anti-biofilm effect, attributed to reactive nitrogen species (RNS), hydroxyl radicals (•OH), and GSH regulation. Bacterial plate counting after four days of treatment showed that SeC@PA MN(+) had a significantly stronger antibacterial effect than SeC@PA NPs(+), demonstrating the advantages of MN-based drug delivery. Unlike conventional methods, MNs penetrate the biofilm barrier, enabling deeper drug distribution, biofilm eradication, oxidative stress reduction, and enhanced wound healing through angiogenesis and collagen deposition. In a 16-day mouse model, SeC@PA MN(+) achieved over 95% wound healing, outperforming the control and other treatments. It also promoted macrophage polarization into the anti-inflammatory M2 phenotype, further accelerating healing. Both in vitro and in vivo studies confirmed that SeC@PA(+) effectively eliminates bacteria, validating the self-amplifying biofilm eradication strategy. The findings revealed that SeC@PA(+) generates active •OH exclusively in the presence of high GSH levels. This indicates that when GSH levels are elevated, SeC@PA(+) contributes to its degradation, increasing RS levels. Conversely, at low GSH levels, it acts as an RS scavenger, enabling bidirectional regulation of reactive species. These findings suggest that SeC@PA dynamically modulates ROS levels, enabling simultaneous biofilm elimination and inflammation reduction without additional therapeutic agents [97].

The difference in conditions within the biofilm itself causes challenges in selecting a stable factor throughout its entire structure. In the study conducted by Cheng et al., micellar nanoparticles were used, designed to move and react within the varied parts of biofilm structures. A micelle is a structure composed of surfactant molecules that form spherical nanostructures capable of dissolving and transporting active molecules, which can effectively handle penetration deep into the biofilm. These nanoparticles contain photocatalysts that can generate nitric oxide (NO) when exposed to red light. The nanoparticles also utilize tertiary amine (TA) groups, which play a stabilizing role and enable the activation of the photocatalysts. Under normoxic conditions in upper layers of biofilms, the TA groups prevent the oxygen-induced photocatalyst quenching, a process where the photocatalyst is deactivated. In the acidic environment found in deeper layers of the biofilm, the TA groups act as proton (H+) acceptors, with the photocatalysts, and promote their deeper penetration into the biofilm. The heterogeneous microenvironments within the biofilm, such as oxygen and pH gradients, result in different properties in the various biofilm layers. The nanoparticles designed in this study are able to adapt to these conditions [98].

### 5.4. Eradication of Fungal Biofilm on the Example of the Problem of Mature Candida spp. Biofilm

Biofilm formation is not only a problem for bacterial infections but also for fungal infections, which are particularly common in diabetic patients. Due to compromised immune systems and high blood sugar levels, diabetic patients are more susceptible to fungal infections that can form biofilms [99]. The incidence of fungal infections in diabetic foot ulcers ranges from 9% to 40.1%. The most common fungal species include *Candida albicans*, *Candida tropicalis*, *Candida parapsilosis*, and *Candida guilliermondii*, followed by *Aspergillus flavus*, *Aspergillus niger*, and *Fusarium* species [100]. The main challenge in treating these infections appears to be the eradication of mature biofilms. Zubara et al., in their in vitro study, focused on examining the inhibition of *Candida albicans* biofilm. *Artemisia vulgaris* L. (Asteraceae) is a medicinal plant used globally to treat various conditions, including diabetes, cancer, and infectious diseases. Given its pharmaceutical significance, their synthesized SnO_2_ nanoparticles use *A. vulgaris* (AvTO-NPs) extract as a stabilizing agent to evaluate their effectiveness in reducing biofilms formed by azole-resistant *Candida albicans* from diabetic foot ulcers (DFUs). At sub-inhibitory concentrations (1/16 × MIC to 1/2 × MIC), AvTO-NPs showed a dose-dependent inhibition, reducing biofilm formation by up to 87.03%. SEM and CLSM analyses revealed disrupted biofilm architecture and decreased adhesion in treated cells. AvTO-NPs also significantly reduced germ tube formation and surface hydrophobicity, key factors in biofilm development. Furthermore, mature biofilms exhibited notable reductions (*p* < 0.05) at concentrations of 1/8 × MIC to 1/2 × MIC, underscoring AvTO-NPs’ potential in combating biofilm-related drug resistance in DFUs [101]. Another important target point for therapy could be the phenomenon of quorum sensing. Tang et al. developed photodynamic nanoparticles (NaYF_4_@NaGdF_4_@PpIX-OC, BaGdF₅@PpIX-OC, and BaGdF₅@SiO_2_-PpIX) activated by NIR to inhibit *Candida albicans* biofilm formation and disrupt mature biofilms. The nanoparticles disrupted quorum sensing by upregulating farnesol and tyrosol through ARO8 and DPP3 expression. Oligo-chitosan (OC)-modified nanoparticles exhibited stronger binding to planktonic cells, effectively inhibiting early-stage biofilm formation, and yeast-to-hyphae transition after laser exposure, and generating the most effective amount of reactive oxygen species (ROS). Among the tested nanoparticles, the smaller (~15 nm) BaGdF₅@SiO_2_-PpIX particles demonstrated superior penetration into mature biofilms despite lower ROS production. However, the destruction efficiency of mature biofilms remained limited to approximately 35%, which is not entirely satisfactory compared to the results obtained for early-stage biofilms. Their findings suggest that ROS generation and nanoparticle binding affinity to planktonic cells play crucial roles in preventing biofilm initiation, while nanoparticle size determines penetration depth in mature biofilms. Notably, the 15 nm *BaGdF₅@SiO_2_-PpIX* nanoparticles achieved a destruction efficiency comparable to the larger *NaYF_4_@NaGdF_4_@PpIX-OC* nanoparticles, which exhibited the highest ROS production. This underscores the necessity of both deep penetration and sufficient ROS generation for effectively eliminating mature fungal biofilms [102].

### 5.5. Conclusion of Possible Biofilm Novel Treatment

In summary, biofilm formation in diabetic patients plays a crucial role in delayed wound healing, as it promotes chronic infections and is highly resistant to standard treatments. Its development is influenced by a complex microbiota, including both aerobic and anaerobic bacteria. Additionally, factors like hyperglycemia, hypoxia, and prolonged infections create conditions that favor biofilm formation and persistence, making eradication significantly more challenging. Modern biofilm-targeting strategies focus on precisely directed methods that not only destroy biofilm structures but also support tissue regeneration. The ability to adapt to the specific conditions of the diabetic foot and the varying microenvironments within biofilms, allowing for deep penetration, appears to be another key factor in combating biofilms. Disrupting quorum sensing mechanisms may also be a valuable therapeutic approach. Together, these strategies offer hope for more effective treatment of diabetic foot ulcers, especially in the face of increasing antibiotic resistance among pathogens. A summary of the most effective therapeutic approaches and the key points on which anti-biofilm therapy focuses has been compiled in Figure 8.

## 6. Materials and Methods

We collected data on diabetic foot ulcers (DFUs) and the potential application of photocatalysis in managing these complications. To conduct our research, we used a systematic search strategy with keywords such as “Photocatalysis in DFU treatment”, “Antimicrobial Photodynamic Therapy (aPDT) in treatment of DFUs”, “Photocatalysis against multi-drug-resistant (MDR) bacteria”, “Nanoparticles in DFU”, “Antimicrobial Photodynamic Therapy in combination with antibiotics”, “Antimicrobial Photodynamic Therapy in combination with other forms of treatment”, and “Photodynamic Therapy against biofilm”.

Our analysis included various protocols for DFU treatment utilizing photocatalysis. Following an evaluation of photocatalysis as a monotherapy, we also examined multiple combination therapies involving photocatalysis to identify approaches that yield the most effective outcomes. Additionally, we investigated the effects of photocatalysis on bacterial biofilms, which represent a significant challenge in DFU management and treatment.

We focused primarily on research articles to obtain a comprehensive and practical overview of the implementation of these methods. Priority was given to studies published in recent years to ensure our findings reflect the most current advances in the field.

## 7. Conclusions

In our review, we provided a comprehensive summary of recent advancements in the application of photocatalysis and photodynamic therapy for the treatment and management of diabetic foot ulcers (DFUs). Our goal was to outline potential clinical strategies adapted to the specific wound microenvironment in diabetic patients.

We analyzed key pathophysiological challenges that hinder therapeutic success in this patient group, including hypoxia, impaired circulation, hyperglycemia, and elevated oxidative stress. These factors collectively pose significant obstacles in the clinical treatment of DFUs.

Our review highlights current therapeutic options available for potential clinical use, such as antimicrobial photodynamic therapy (aPDT), liposomal photocatalytic carriers, and nanoparticles, either as standalone treatments or as part of combination therapies with agents such as antibiotics, lysozyme, and bacteriophage-derived enzymes, aiming not only to control infection progression but also to eradicate it, prevent recurrence, and, most importantly, facilitate effective tissue regeneration. In APDT and photocatalysis for diabetic wounds, NIR light enables deeper tissue penetration, while UV light, despite shallower reach, offers stronger photochemical effects. Choosing the right wavelength is key for effective therapy.

Additionally, we emphasized the critical role of biofilm elimination in DFUs, as persistent biofilms significantly contribute to chronic infection and treatment resistance. Their effective removal is a key component of contemporary therapeutic strategies aimed at eradicating the underlying pathogenic microorganisms.

## Figures and Tables

**Figure 1 molecules-30-02323-f001:**
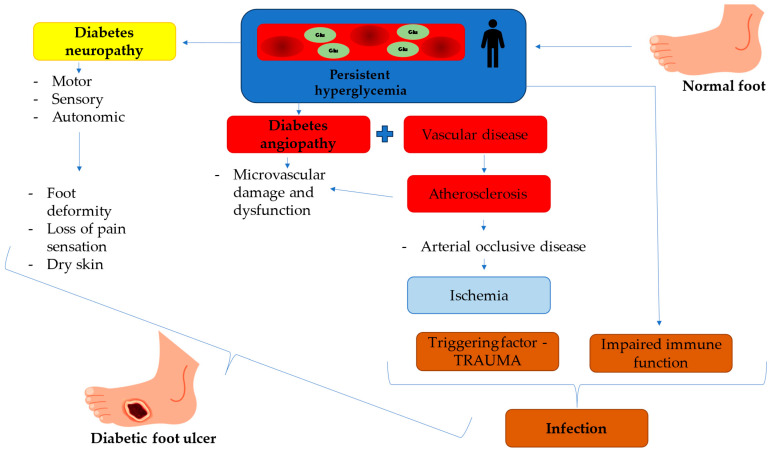
Pathophysiology of diabetic foot ulcers.

**Figure 2 molecules-30-02323-f002:**
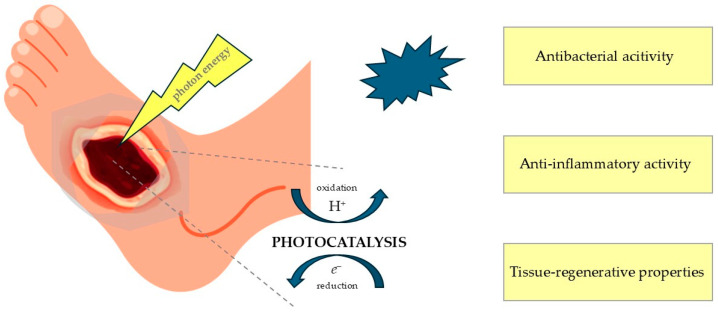
Fundamental benefits of photocatalysis in the treatment of diabetic foot ulcers (DFUs).

**Figure 3 molecules-30-02323-f003:**
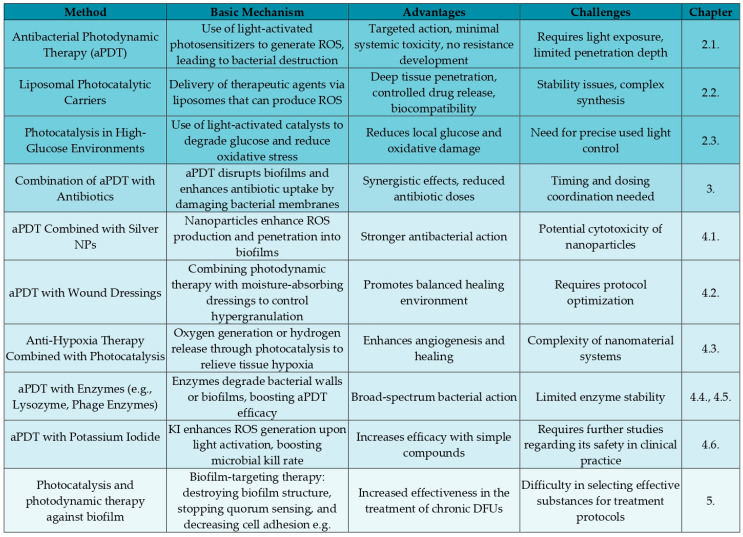
Summary of photocatalysis and photodynamic therapy methods in diabetic foot ulcer (DFU) care discussed in this review, along with a comparison of the key features of individual approaches.

**Figure 4 molecules-30-02323-f004:**
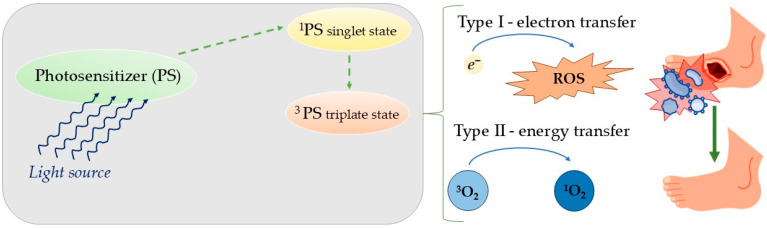
Mechanism of action of antibacterial photodynamic therapy (aPDT).

**Figure 5 molecules-30-02323-f005:**
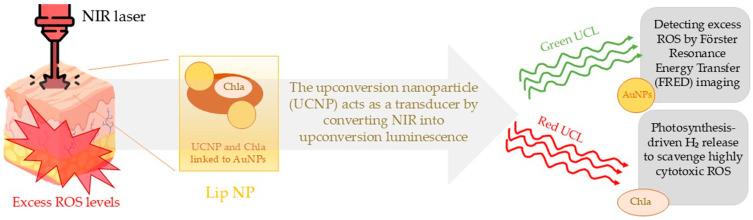
Mechanism of action of a photo-driven H_2_-releasing liposomal nanoplatform (Lip NP) composed of an upconversion nanoparticle (UCNP) linked to gold nanoparticles (AuNPs) through an ROS-sensitive connector and the usage of near-infrared (NIR) light laser.

**Figure 6 molecules-30-02323-f006:**
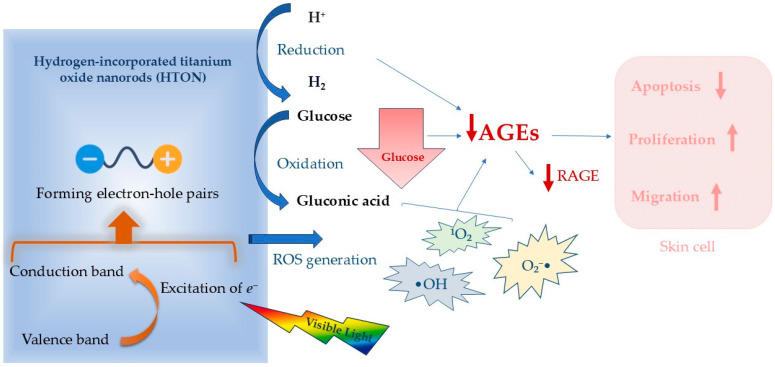
Schematic illustration of the mechanism of action of hydrogen-incorporated titanium oxide nanorods in combination with visible light on skin cells.

**Figure 7 molecules-30-02323-f007:**
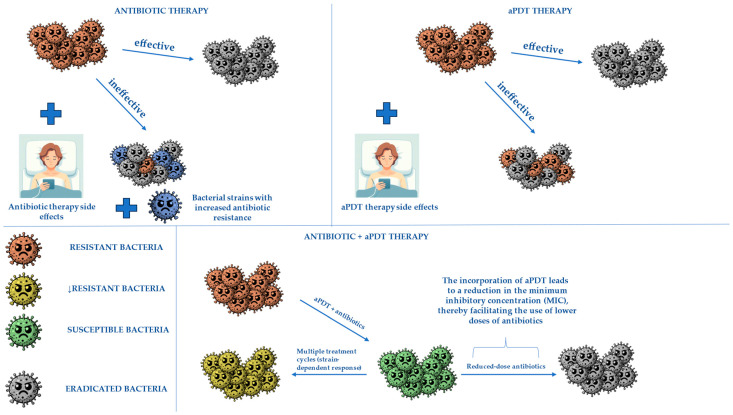
The additive effect of combining antibiotic therapy with antimicrobial photodynamic therapy.

**Figure 8 molecules-30-02323-f008:**
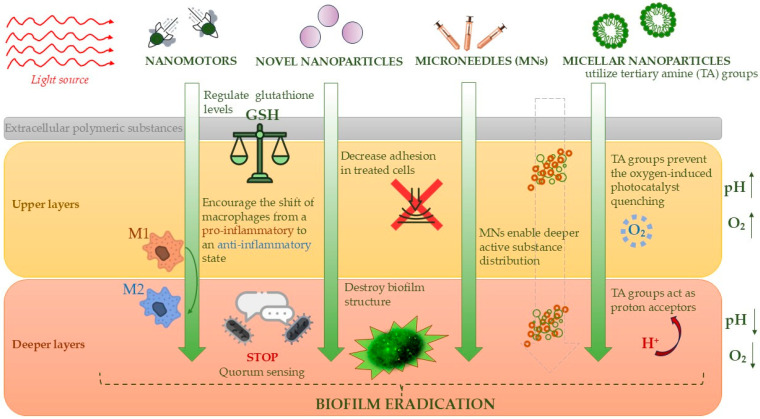
Comparison of useful methods for combating biofilm structure: nanomotors, novel nanoparticles, microneedles, micellar nanoparticles, along with a schematic representation of their mechanism of action.

**Table 1 molecules-30-02323-t001:** Comparison of structural features and susceptibility to photocatalysis of Gram-positive and Gram-negative bacteria.

Feature	Gram-Positive Bacteria	Gram-Negative Bacteria
**Cell Wall Thickness**	20–80 nm	10 nm
**Peptidoglycan Content**	>50%	10–20%
**Lipid and Lipoprotein Content**	0–3%	58%
**Presence of Lipopolysaccharides (LPSs)**	Absent	13%
**Permeability to Reactive Oxygen Species (ROS)**	High—the cell wall is porous, facilitating ROS penetration	Low—the outer lipid membrane restricts ROS access
**Effectiveness of Photocatalysis**	Higher—ROS easily penetrate, causing damage to proteins, lipids, and DNA	Lower—requires outer membrane damage first, demanding longer exposure and higher energy
**Effect of ROS Damage**	Rapid loss of proteins and K^+^ ions, DNA damage, enzyme denaturation	Slow outer membrane damage first, then cytoplasmic and genetic material effects
**Additional Defense Mechanisms**	Some bacteria produce endospores, biofilm, or a polysaccharide layer	LPS and biofilms provide protection against photocatalysis

## Data Availability

No new data were created or analyzed in this study.

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
