# Peer review of "Photocatalysis and Photodynamic Therapy in Diabetic Foot Ulcers (DFUs) Care: A Novel Approach to Infection Control and Tissue Regeneration"

_molecules, 2025, doi:10.3390/molecules30112323_

Round 1

Reviewer 1 Report

Comments and Suggestions for Authors

Comments:

This review comprehensively explores the application of photocatalysis and photodynamic therapy (PDT) in diabetic foot ulcer (DFU) management, covering advanced strategies such as antibacterial PDT (aPDT), liposomal photocatalytic carriers, nanoparticle-based combinatorial therapies, and biofilm eradication.Here are some suggestions:

  1. The article mentioned that Rh/Agâ‚‚MoOâ‚„ relieved local hypoxia through photothermal effect (Li et al. 2024), but did not specifically explain how it regulates the HIF-1α/VEGF pathway. It is recommended to supplement relevant molecular mechanism experiments or literature support.
  2. The mechanism of photocatalytic reduction of local glucose concentration (such as HTON of Chen et al.) needs to be further explained in terms of its specific association with the inhibition of advanced glycation end products (AGEs).
  3. Long-term biosafety data for nanomaterials (such as Au-Ag core-shell nanoparticles) are missing. It is recommended to supplement cytotoxicity and in vivo metabolic pathway studies (such as effects on liver and kidney function).
  4. Subdivide Section 6 (“Biofilm Eradication”) into bacterial vs. fungal biofilms for clarity.
  5. Strengthen the Discussion (Section 8) by explicitly addressing limitations (e.g., light penetration depth, NIR vs. UV applicability).
  6. Update citations (e.g., Peppa et al., 2009) to reflect recent advances (last 5 years).
  7. The format of the full text of some chemical formulas is not uniform (for example: line 724, line 727, etc., Ag2MoO4).

Author Response

Thank you for reviewing our work! Here are our answers to the suggestions made:

  1.  The article mentioned that Rh/Agâ‚‚MoOâ‚„ relieved local hypoxia through photothermal effect (Li et al. 2024), but did not specifically explain how it regulates the HIF-1α/VEGF pathway. It is recommended to supplement relevant molecular mechanism experiments or literature support. –

Thank You for this question. Corrected. The presentation of the cause of the therapy's success in counteracting hypoxia and its effects has been refined.

2. The mechanism of photocatalytic reduction of local glucose concentration (such as HTON of Chen et al.) needs to be further explained in terms of its specific association with the inhibition of advanced glycation end products (AGEs). –

Thank You for suggestion. Corrected in text and added to Figure 5.

3. Long-term biosafety data for nanomaterials (such as Au-Ag core-shell nanoparticles) are missing. It is recommended to supplement cytotoxicity and in vivo metabolic pathway studies (such as effects on liver and kidney function). –

Thank You for this question. This section has been revised, additional information about silver nanoparticles safety and risks has been added, with a focus on potential adverse effects on the liver and kidneys, along with the inclusion of important citations.

4. Subdivide Section 6 (“Biofilm Eradication”) into bacterial vs. fungal biofilms for clarity. –

 Thank you for this important suggestion. Corrected.

5. Strengthen the Discussion (Section 8) by explicitly addressing limitations (e.g., light penetration depth, NIR vs. UV applicability). –

Thank you for this suggestion. Corrected.

6. Update citations (e.g., Peppa et al., 2009) to reflect recent advances (last 5 years).

Corrected. Thank you for this suggestion. Updated and added new citations.

7. The format of the full text of some chemical formulas is not uniform (for example: line 724, line 727, etc., Ag2MoO4). –

Thank you for this suggestion. Corrected.

Thank you for your contribution to our manuscript. Your work helps us create higher quality work.

Reviewer 2 Report

Comments and Suggestions for Authors
  1. This non-toxic inorganic salt, can counteract this antioxidant effect by releasing free iodine upon reacting with the 1O2 generated through red light activation of IRDye 700DX. What does this mean?How could iodine counteract this antioxidant effect?
  2. Representative figures are scarce, particularly those illustrating therapeutic mechanisms and pathways.
  3. The overall logic of the review is somewhat disorganized. Although it claims to focus on photocatalysts and photodynamic therapy, it lacks a systematic introduction to the types, properties, and mechanisms of photocatalysts, while devoting excessive attention to other types of therapeutic agents.
  4. The cited literature needs to be updated and should include more influential references.
  5. A comprehensive schematic diagram should be added to clearly convey the core content of this review.

Author Response

Thank you for reviewing our work! Here are our answers to the suggestions made:

  1. This non-toxic inorganic salt, can counteract this antioxidant effect by releasing free iodine upon reacting with the 1O2 generated through red light activation of IRDye 700DX. What does this mean?How could iodine counteract this antioxidant effect?

Thank You for this suggestion. This part of text has been improved in order to better explain the application of KI.

2. Representative figures are scarce, particularly those illustrating therapeutic mechanisms and pathways.

Thank You for this suggestion. Several figures have been adjusted and refined without compromising the clarity we intended to preserve.

3. The overall logic of the review is somewhat disorganized. Although it claims to focus on photocatalysts and photodynamic therapy, it lacks a systematic introduction to the types, properties, and mechanisms of photocatalysts, while devoting excessive attention to other types of therapeutic agents.

Thank You for this suggestion. This section has been expanded with an introductory paragraph on photocatalysts. We focused on those photocatalysts that, according to the current scientific literature, show real potential for application in the treatment of diabetic foot ulcers (DFUs). Each photocatalyst has been characterized, and for each, we also included a discussion of their safety profile. Our attention was directed specifically towards photocatalysts relevant to DFU treatment in order to maintain focus on the main objective of our study, which is their potential in addressing DFUs. We did not reduce the information regarding other therapeutic agents, as we believe they represent an important part of our work and further enrich the discussion on the therapeutic potential of combinations with photocatalysis and antimicrobial photodynamic therapy (aPDT).

4. The cited literature needs to be updated and should include more influential references.

Thank you for this suggestion. Updated and added new citations.

5. A comprehensive schematic diagram should be added to clearly convey the core content of this review.

Thank you for this suggestion. Added at the beginning of the review.

Thank you for your contribution to our manuscript. Your work helps us create higher quality work.

Round 2

Reviewer 1 Report

Comments and Suggestions for Authors

This is a revised version of a previously submitted manuscript. Overall the authors have addressed the questions raised by the reviewers. I would suggest acceptance without further changes.